# Classification of Participants Based on Increase–Decrease Rate Model of Reaction Time to Personality Trait Words

**DOI:** 10.3390/bs9120155

**Published:** 2019-12-14

**Authors:** Kouhei Matsuda, Emi Sato

**Affiliations:** 1Department of Human Welfare, Tohoku Bunkyo College, Yamagata 990-2316, Japan; 2Department of Business Administration, Tokyo Fuji University, Tokyo 169-0075, Japan; karen@ff.iij4u.or.jp

**Keywords:** personality, reaction time, trait words, variation rate, cluster analysis

## Abstract

In this experiment, we tried to measure personality by reaction time (RT) to stimuli of personality trait words. There were interindividual and intraindividual differences in the factors that caused the reaction time to fluctuate. The intraindividual differences for personality trait words were caused by changes due to circumstances for the same participant. The increased stimulus reaction time (sRT) model for simple reaction time was used as an index to indicate personality traits. As a result, participants could be classified into two major hierarchical clusters. The participants in Cluster 1 showed innovative dominance. The participants in Cluster 2 were obedient and conservative. The independent variable was measured by the physiological index using sRT for classify the participants. Participants in Cluster 2 had a reduced stress response to the experiment and showed a tendency to be compliant. Moreover, immediately after the RT measurement session with a laptop computer started, participants in Cluster 1 showed decreased HEG and increased amylase values and had a somewhat negative attitude. The physiological dependent variable were measured by using salivary amylase and hemoencephalography (HEG). And, the psychological dependent variable was the Big Five personality inventory. All of them ware using to verify the participant’s classification. Participants in Cluster 2 had significantly higher conscientiousness than those in Cluster 1. Therefore, we suggest that it is possible to classify personality traits from RT by using sRT based on intraindividual differences.

## 1. Introduction

Reaction time (RT) is an approximate value that signifies a complex sum of biological responses and psychological effects [1]. It is difficult to identify intraindividual differences in RT, and they are often considered errors in measurement. In this study, we assume that there are interindividual differences in RT in response to stimuli, and examine the increased stimulus reaction time (sRT) model for the intraindividual variation rate. Using this sRT model, we examine whether it is possible to identify certain personality traits in terms of RT with regard to mental performance of personality self-rating.

The RT for certain tasks is a psychological index and indicator of performance that shows the amount and complexity of psychological processing. The purpose of this experiment is to identify certain personality traits in terms of RT with regard to mental performance in assessing one’s own personality. We measure RT and cerebral blood flow during three rating sessions on the computer: simple response, self-rating reaction with words by term, and self-rating reaction with text by sentence. Individual differences are cited in cognitive psychology as a determining factor in simple RT [2]. This individual difference is a problem, and it is difficult to examine personality traits that may differ between individuals in a performance task. However, standard deviations of RT are typically viewed as errors, but such errors (from trial to trial) are individual differences [3,4].

We assume that one factor in the varying rate of RT is fatigue and impulse by repeatedly responding to many items of the self-rating personality test, and that this factor causes variation in the speed of RT. That is, we consider the influence of mental load by repeated stimuli and reactions. Therefore, focusing on the amount of change in the individual’s response to the stimulus, we examine the increased stimulus reaction time (sRT) model, which measures the rate of the increase of response time of each individual to stimulus words.

The sRT model is based on the participant’s response time to a stimulus in a simple response session. In the simple response session, we measure individual differences in cerebral blood flow during stimulation and reaction. Given certain mental performance, the individual response is greater than the average response. Therefore, we set up a simple response session for each individual to match the auditory and visual stimulus in the experiment, and consider it as each individual’s average response time to the stimulus. Next, with regard to the mental performance of self-rating one’s personality, we measure the RT in two sessions, a self-rating by term session and a self-rating by sentence session.

The mean of the simple response session is defined as the intraindividual difference in the response to the stimulus, and we consider the time varying from that standard to be the time for each individual’s self-rating of personality. Then, we consider that it is possible to identify personality traits common among individuals by dividing the RT of performance by the mean of the simple response session of each individual. In a study that examined the relationship between scales from the Big Five personality inventory and RT using the sRT model, it was found that extroverted persons varied in their personality self-rating performance [5]. This suggests that extroverted persons repeatedly responded to a number of items using a computer, thereby increasing the amount of change for each individual’s performance. Thus, this study reveals that it is possible to indicate quantitative data about an individual by considering intraindividual differences in RT during personality self-rating.

Therefore, in this research, we examine whether it is possible to identify a certain personality feature as the mental performance of personality self-rating by using the sRT model [6]. Each sRT is calculated by the following equation from the intraindividual means of simple RT and self-rating RT for each trait word:(1)$${sRT}_{k}=\frac{RT{}_{k}-mrt{}_{k}}{RT{}_{k}}$$
where *mrt_k_* is the intraindividual mean of simple reaction time *RT_k_* in Session 1 of each trait word *k*.

The sRT is calculated for each participant by multiplying each of the factors of the Big Five by four trait words, so the sRT per participant is 20 traits. For these, intraindividual differences are used to detect and classify interindividual differences, and to examine the relationship with each dependent variable. The index measures the dependent variables by using hemoencephalography (HEG) and salivary amylase as physiological responses, and the scale values of the Big Five as a psychological response.

## 2. Materials and Methods

### 2.1. Participants

The participants included 22 Japanese university students (13 men, 9 women) aged 19 to 22 years. One woman was excluded due to incomplete data, therefore data for 21 persons were used for our analysis. All participants were academic volunteers. We paid 1000 JPY to each participant as a slight reward.

### 2.2. Experimental Period

The experimental period extended from May through to December 2014.

### 2.3. Informed Consent to the Participants

The experimenter fully explained the content of the experiment to the participants and answered their questions. In addition, we obtained written consent from the participants before the experiment. These procedures were reviewed by the ethics committee of each author’s affiliation and judged appropriate.

### 2.4. Equipment

The equipment included a laptop computer (Dell Vostro 3360) and a ProComp TM7500 vital signs monitor (Thought Technology Ltd., Canada) to measure the prefrontal cortex according to hemoencephalography (HEG), heart rate (HR), and electroencephalography (EEG) (electrodes of international 10-20 system: F3, F4). To present stimuli during the three sessions of the self-rating task, we installed E-prime 2.0 on the computer. The degree of stress during the self-rating task was measured using a salivary amylase monitor (NIPRO; 27B1 × 00045000073).

### 2.5. Personality Inventory and Experiment Stimulus

In self-rating their personality with paper and pen, participants rated the bipolar scale construction of the Big Five inventory for Japan [7], the behavioral inhibition system and behavioral activation system (BIS/BAS) scale for Japan and the self-control scale [8,9]. In self-rating personality with the computer, we selected terms that would be familiar to university students for each of the personality traits in Table 1, referring to a manual of the Big Five for Japan. These 20 personality trait terms were set as visual stimuli in computer image files for display on the screen. Auditory stimuli of these 20 traits were then recorded using a male voice (700 ms duration).

### 2.6. Procedure

To measure frontal cerebral blood flow during personality rating tasks, we set six conditions, shown in Table 2. Two questionnaire conditions and three rating session conditions were set on the computer, and the closed-eyes condition was set as the criterion for the individual. After explaining the experiment to the participant, the experimenter set up the electrocardiogram, the frontal EEG (F3, F4), and the HEG using the vital signs monitor. To confirm the installation of the equipment, the experimenter asked the participant to read a newspaper and confirmed that frontal cerebral blood flow could be measured using HEG. After confirming the installation of the equipment, the experimenter began to measure self-rating conditions of brain activity.

In the Big Five condition, to measure cerebral blood flow using HEG, each participant completed the Big Five using pen and paper. After completing the inventory, the participant’s salivary amylase was measured for 30 s to determine the degree of stress. To measure frontal cerebral blood flow during the stabilization period, we used the closed-eyes condition. We measured cerebral blood flow during three rating sessions on the computer: simple response, self-rating by term, and self-rating by sentence.

In the simple response session, we measured individual differences in cerebral blood flow during stimulation and reaction. After displaying a gaze point (+) on the computer for 500 ms, we displayed a black dot (●) for 700 ms. upon seeing the black dot, the participant heard an audio stimulus via headphones. A visual stimulus was displayed on the next screen (max = 1800 ms). If the audio and visual stimuli matched, the participant pressed “○.” If the terms did not match, the participant pressed “×”. After the participant pressed a key, that trial was finished (masking). The visual and audio stimuli were used for the 20 personality terms in Table 1, and the trial was assigned randomly for each participant. In a practice session, the participant practiced 15 times in the simple response session (three words × 5). After the participant got used to this operation, we performed 200 trials in the simple response session (20 terms × 10), as shown in Figure 1.

In the self-rating by term session, we measured the self-rating of the stimulus terms (Table 1) to measure cerebral blood flow during the task with only trait terms (Figure 1). After displaying a gaze point (+) for 500 ms and a black dot (●) for 700 ms, we randomly displayed a trait term as a visual stimulus on the next screen. In this session, a black dot (●) screen was not coupled with an audio stimulus. When participants observed a trait term that they thought applied to their own personality, they pressed “○.” Conversely, when participants observed a trait term that they did not think applied to their own personality, they pressed “×.” The trial was then finished (masking). Participants performed 100 trials (20 terms × 5).

In the self-rating by sentence session, we added “Are you …” at the beginning of the sentences used in the previous by term session. After silently displaying a gaze point (+) for 500 ms and a black dot (●) for 700 ms, we randomly displayed an “Are you …” sentence using a trait term on the next screen. For example, a participant observed the sentence “Are you kind?” and pressed “○” for a positive response and “×” for a negative response. Each participant performed 60 trials (20 terms × 3).

In the questionnaire answering (QA) condition, after three sessions on the computer, the participant completed the BIS/BAS and self-control scales. After another 30 s salivary amylase measurement, the experimenter removed the experimental equipment, and the experiment was completed.

### 2.7. Data Analysis

The sRTs of the 20 words of each trait per participant were determined according to Equation (1). Based on these sRTs, we calculated the squared Mahalanobis distance as the similarity index between participants. The nearest-neighbor chain algorithm can be used to find the same clustering defined by Ward’s clustering method. Therefore, we performed hierarchical cluster analysis by Ward’s method. The differences between classified clusters were compared for HEG, salivary amylase, and the Big Five as dependent variables. Only in this analysis, the RT by sentence was saved for further study.

### 2.8. Ethical Considerations

All participants gave their informed consent for inclusion before participating in the experiment. The study was conducted in accordance with the Declaration of Helsinki, and the protocol was approved by the ethics code of JSPS KAKENHI, grant number 24530846, by the Japanese Society for the Promotion of Science.

## 3. Results

### 3.1. Cluster Analysis

The participants were 21 university students, 13 men and 8 women. They were classified by hierarchical cluster analysis based on Ward’s method using sRTs obtained from 20 trait words. The index of the distance between participants was the squared Mahalanobis distance. A dendrogram was obtained from this cluster analysis, shown in Figure 2. The dendrogram shows that the participants were classified into two major clusters. The agglomerative coefficient of these two clusters was 1.33 times the ratio of standard deviation.

### 3.2. Cluster Classified Reaction Time by Personality Trait Term

As a result of cluster analysis, Cluster 1 was composed of 10 participants and Cluster 2 was composed of 11 participants, as shown in Table 3. We performed simple RT by two-way ANOVA with the clusters and the Big Five personality factors as independent variables. Significant differences in the main effects were observed between clusters (*F(1, 30) = 48.28, p < 0.01*), and between the Big Five factors (*F(4, 30) = 11.31, p < 0.01*). However, no interaction between clusters and factors was observed (*F(4, 30) = 0.11, ns*). Similarly, for self-rating RT, a main effect between clusters was observed (*F(1, 30) = 19.43, p < 0.01*), and between the Big Five factors (*F(4, 30) = 11.31, p < 0.01*) was not observed (*F(4, 30) = 0.28, ns*), and no interaction between clusters and factors was observed (*F(4, 30) = 0.67, ns*). In addition, since sRT is a classification variable, the ANOVA of sRT was not of value to examine the statistical tests.

### 3.3. Clustered Physiological Index

As a result of cluster analysis using sRT as a classification variable, participants were classified into two major clusters. Significant differences were observed in salivary amylase values after the task (*t(19) = 2.44, p < 0.05*). The salivary amylase value (in U/L) in Cluster 1 was high, with a mean of 63.8, and standard deviation was also high at 59.9, and in Cluster 2 it was low, with a mean of 18.7 and standard deviation of 13.1. Participants in Cluster 1 had increased stress responses and those in Cluster 2 had decreased responses after the computer session.

HEG was analyzed using intraindividual variation with the closed-eyes condition as a baseline. In HEG, a significant difference was found in the simple reaction time session (*t(19) = 1.73, ns*). However, it had a significant difference tendency. The participants in Cluster 1 had decreased cerebral blood flow (93.3%), and those in Cluster 2 had no change.

Classification by sRT was similar to the result of the HEG classification, in which the two clusters were classified into the main two and lower five categories [10] (see Figure 3).

### 3.4. Clustered Psychological Index

In cluster comparison by the Big Five, a significant difference among clusters was observed only for conscientiousness (*t(19) = 1.97, ns*). However, it had a significant difference tendency. In Cluster 1, conscientiousness had a mean of 3.60 and a standard deviation of 3.17. In Cluster 2, conscientiousness had a mean of 6.00 and a standard deviation of 2.41. It was difficult to detect the difference due to sRT, which was the behavior characteristic from the psychological index by the questionnaire (see Figure 4).

## 4. Discussion

Cluster analysis by the increased stimulus reaction time (sRT) model was classified with a high agglomerative coefficient of 1.33 σ. From this result, the variation rate of intraindividual reaction times due to conditions was effective in detecting interindividual differences. As shown in Table 3, the participants of Cluster 2 clearly had an increased judgment time with the trait words active, restrained, tightwad, sedate, and intelligent. RT measured as time to judge negative stimulus words increased [11]. The participants in Cluster 1 showed innovative dominance. The participants in Cluster 2 were obedient and conscientious.

As shown in Figure 3 participants in Cluster 2 had a reduced stress response due to the experiments according to salivary amylase values, and showed a tendency to be compliant. Similarly, immediately after the computer session started, the HEG of participants in Cluster 1 decreased and they had a somewhat negative attitude. As shown in Figure 4 participants in Cluster 2 had significantly higher Big Five conscientiousness than those in Cluster 1. The variation of these dependent variables supported the results of cluster analysis by sRT.

The personality inspection by the questionnaire was to some extent self-reporting [12] and had a lack of internal consistency [13]. The results of the questionnaire were not relevant enough for criterion-related validity, and arbitrary answers were possible. This RT-based personality trait classification experiment was measured by behavioral indicators. The results show the possibility of applying it to a personality test with objectivity. This personality measurement method was considered effective in situations where respondent self-bias occurs.

This study was based on HEG. This study should be replicated with other novel neuroimaging modalities. Functional near-infrared spectroscopy (fNIRS) is noninvasive neuroimaging that maps the functions of the cerebral cortex by measuring hemodynamics and is cost-effective [14]. fNIRS has the advantage of being readily translated to clinical use as it is more cost-effective than functional magnetic resonance imaging (fMRI) [15]. Future research should study RT by fNIRS in clinical populations.

## 5. Conclusions

As in previous studies, RTs were indicators of generalized individual differences, and variations within individuals were regarded as errors. The within-individual difference of self-rating RT with regard to simple reaction RT was indexed as sRT according to the increased stimulus reaction time model. The results derived two unique clusters by sRT. Therefore, we suggest the possibility of classifying personality traits from RT by using sRT based on intraindividual differences.

## Figures and Tables

**Figure 1 behavsci-09-00155-f001:**
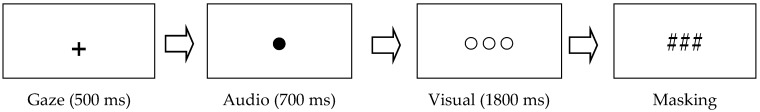
Sequence of stimulation in simple response session (Session 1).

**Figure 2 behavsci-09-00155-f002:**
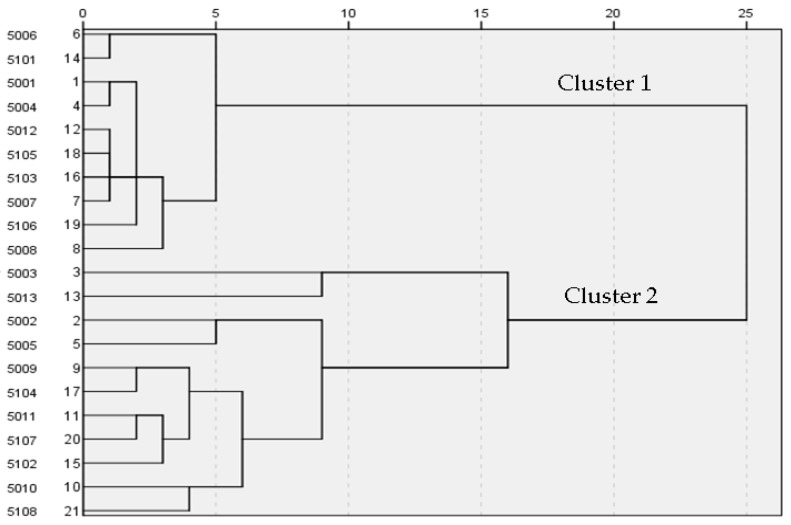
Dendrogram of 21 participants’ words classified by stimulus reaction time (sRT). Vertical axis shows participant identifications and corresponding numbers, and horizontal axis shows the similarity index.

**Figure 3 behavsci-09-00155-f003:**
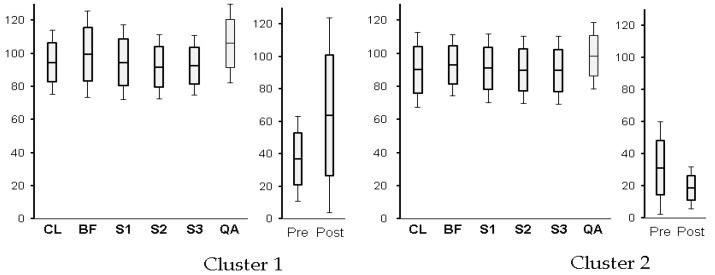
HEG and salivary amylase values as clustered physiological indices. CL: eyes closed for 3 min as a rest time; BF: self-rating with the Big Five inventory before computer sessions; S1: simple response to trait words (Session 1) on computer; S2: personality self-rating by term (Session 2); S3: personality self-rating by sentence (Session 3); QA: self-rating with questionnaire after computer sessions; Pre: salivary amylase before computer session; Post: salivary amylase after all sessions.

**Figure 4 behavsci-09-00155-f004:**
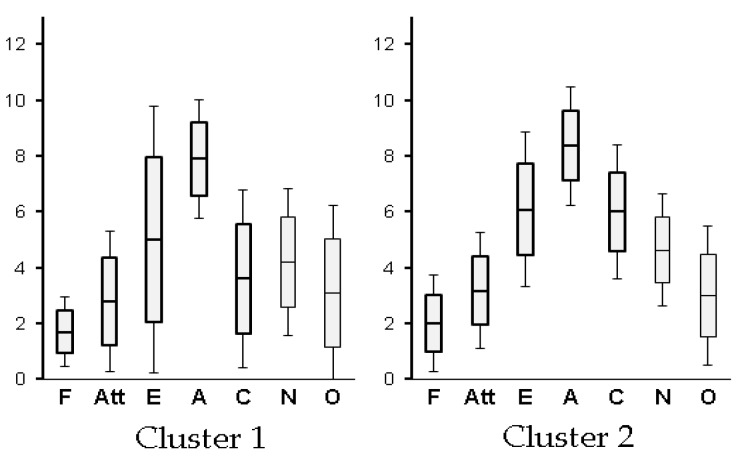
The Big Five personality inventory scales as clustered psychological indices. CL: eyes closed for 3 min as a rest time; BF: self-rating with the Big Five inventory before computer sessions; S1: simple response to trait words (Session 1) on computer; S2: personality self-rating by term (Session 2) on computer; S3: personality self-rating by sentence (Session 3) on computer; QA: self-rating with questionnaire after computer sessions; Pre: salivary amylase before computer sessions; Post: salivary amylase after all sessions.

**Table 1 behavsci-09-00155-t001:** Stimulus terms of personality trait words used in the experiment.

Trait Factor		Stimulus Terms/Variable Names			
Big Five Factors	Extroversion	active	sociable	passive	restrained
	Agreeableness	kindly	affable	headstrong	tightwad
	Conscientiousness	capable	conscientious	sloppy	unreliable
	Neuroticism	easygoing	sedate	irascibleness	worrier
	Openness to experience	intelligent	clever	conservative	naïveté
Practice Stimulus Terms		sincere	amenable	philosophical	-

**Table 2 behavsci-09-00155-t002:** Experimental sessions and measurement conditions: hemoencephalography (HEG).

Session	HEG Conditions	Contents
Closed-eyes condition	CL	Eyes closed for 3 min as a rest time
Questionnaire condition	BF	Self-rating with Big Five inventory before computer session
Amylase 1	Pre	Degree of stress before computer session
Computer condition	S1	Session 1: Simple response to trait words session
	S2	Session 2: Personality self-rating by term session
	S3	Session 3: Personality self-rating by sentence session
Questionnaire answering condition	QA	Self-rating with questionnaire after computer session
Amylase 2	Post	Degree of stress after all sessions

**Table 3 behavsci-09-00155-t003:** Classified RT and sRT of each trait word by cluster analysis of participants.

Trait Words	Cluster 1						Cluster 2					
	Simple		Self-rating		sRT		Simple	Self-rating			sRT	
	*mean*	*sd.*	*mean*	*sd.*	*mean*	*sd.*	*mean*	*sd.*	*mean*	*sd.*	*mean*	*sd.*
Extroversion												
active	515.1	*107.1*	761.3	*157.9*	0.861	*0.384*	574.1	*99.7*	1163.0	*493.0*	0.900	*0.477*
sociable	500.2	*119.5*	767.1	*210.4*	0.604	*0.283*	539.5	*120.1*	848.9	*233.2*	0.658	*0.466*
passive	512.9	*130.5*	996.0	*420.8*	0.985	*0.409*	566.7	*134.7*	1177.0	*377.7*	0.740	*0.494*
restrained	505.6	*123.8*	894.2	*199.4*	0.742	*0.316*	530.6	*112.6*	1080.1	*252.1*	0.841	*0.679*
Agreeableness												
kindly	541.2	*104.0*	851.1	*227.2*	0.465	*0.347*	583.5	*115.6*	840.1	*127.6*	1.075	*1.107*
affable	536.4	*117.4*	811.2	*134.6*	0.432	*0.220*	592.6	*115.0*	1053.9	*368.3*	0.513	*0.566*
headstrong	500.9	*90.4*	878.3	*201.4*	0.862	*0.699*	544.8	*101.5*	1016.6	*192.3*	1.076	*0.750*
tightwad	535.3	*124.1*	908.9	*251.7*	0.690	*0.333*	580.8	*111.4*	1337.0	*396.8*	0.884	*0.446*
Conscientiousness												
capable	580.9	*123.3*	985.6	*247.8*	0.580	*0.186*	651.6	*135.9*	1076.5	*207.3*	0.457	*0.173*
conscientious	564.7	*99.6*	839.4	*129.4*	0.546	*0.257*	604.4	*130.8*	942.9	*238.8*	0.867	*0.707*
sloppy	617.6	*144.1*	1073.4	*336.0*	0.670	*0.348*	667.4	*115.3*	990.4	*252.1*	0.800	*0.467*
unreliable	551.5	*110.8*	935.6	*271.8*	0.708	*0.344*	598.2	*115.7*	1046.9	*373.0*	1.351	*0.749*
Neuroticism												
easygoing	533.7	*98.5*	886.4	*239.2*	0.691	*0.477*	592.8	*150.9*	886.9	*190.7*	0.556	*0.364*
sedate	521.5	*89.7*	904.1	*215.0*	0.759	*0.550*	573.1	*131.2*	1267.5	*544.5*	1.254	*1.039*
irascibleness	509.5	*91.8*	930.9	*299.0*	0.737	*0.451*	567.3	*126.6*	1014.1	*378.7*	0.808	*0.751*
worrier	534.7	*137.3*	808.7	*148.5*	0.523	*0.128*	572.8	*125.6*	896.5	*170.8*	0.578	*0.357*
Openness to experience												
intelligent	555.2	*110.9*	841.1	*188.6*	0.602	*0.362*	637.8	*139.2*	1061.1	*227.8*	0.856	*0.449*
clever	530.7	*85.7*	819.1	*181.2*	0.545	*0.285*	549.4	*75.3*	987.3	*208.1*	0.758	*0.532*
conservative	533.4	*114.4*	754.0	*98.4*	0.435	*0.173*	587.3	*142.0*	1088.1	*370.4*	0.929	*0.692*
naïveté	503.9	*83.1*	889.3	*270.6*	0.716	*0.604*	582.3	*116.9*	977.1	*216.7*	0.698	*0.361*

Note: Trait words were in accordance with the Big Five in Table 1. Reaction time (RT) was measured in milliseconds. sRT coefficients were calculated as the ratio of RT by Equation (1).

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
