# Peer review of "Classification of Participants Based on Increase–Decrease Rate Model of Reaction Time to Personality Trait Words"

_behavsci, 2019, doi:10.3390/bs9120155_

Round 1

Reviewer 1 Report

In Abstract:

1) Please, revise whether it is correct "obedient and obedient": "The participants in cluster 2 were
17 obedient and obedient.

2) In what th ecluster 1 has decreased, please, specify since it is not clear: "Moreover immediately after the PC
20 session started, the cluster 1 decreased and was somewhat negative attitude. " and idem: less respons in what? further in Abstract:

"The cluster 2 had less stress response due to
19 experiments and they showed a tendency to be compliant."

and "Moreover immediately after the PC
20 session started, the cluster 1 decreased and was somewhat negative attitude. "

Please, write, theses parts of Abstract for a more comperehensive information of resutls of the study.

May be the authosrd need to pass the correction or translation with a native speaker for it to improve.

METHODS

1) How the students were recuited, on which base (voluntary or any kind of compensation), and what were criterias of their inclusion/exclusion to the study?

2) Somewhere in the beggining it is good also to mention about Japanese "bipolar" scales since some readers may not know about it and imagine the Big Five they accostomed to use. It can be introduced here, for example, before "scale":

"In self-rating personality with paper and pen, participants rated the scale construction of the
101 BigFive Inventory for Japanese [6.."

3) Though some abbreviations are clear for many people, still, it is good to provide the desclosure of their abbreviations when it is mentioned for the first time in the manuscript: HEG, HR, EEG:

"93 2.4 Equipment
94 The equipment included a laptop computer (Dell-Vostro 3360), vital monitor ProComp TM7500
95 (Thought Technology Ltd, Canada) to measure the prefrontal cortex according to HEG, HR, EEG (F3, F4). "

RESULTS

1) Which "factor"?, please, specify:

185 p.<.01). However, no interaction between cluster and factor was observed (F(4, 30)=0.11, ns.).

Or do you mean all factors?

2) In Table 3, it will be better if you specify (or even give titles) to each cluster with refference on RT and/or other parameters 

3) Please, specify, which "dependent variable" (line 193)? in:

192 3.3 Clustered physiological index
193 As a result from dependent variable, participants were classified into two major clusters. The...

4) Please, give the disclosure for abbreviations given on Figures 4 & 5 in the Notes below.

5) At the Figure 3, please, specify in Notes what the numbers in two vertical colomns stand for? (subjects ids and ???)

6) Why the authors considered p<.10 as statistically significant level or there is a typing error:

208 In cluster comparison by BigFive, significant differences among clusters were observed only
209 in conscientiousness (t(19)=1.97, p.<.10). I

7) The small sample size should be explained as Limitations of study.

The authors can check to apply the bootstrapping of the sample prior or incorporated within analyses to reduce the variance of samples of both clusters to see if there any changes in the results (recommended). 

DISCUSSION

Study Limitations should be mentioned as sample size (though for physiological measurements less sample size is used compared to other psychological studies usually due to procedure).

Also the direction for future studies it is good to indicate in the Discussion.

CONCLUSIONS

If I undestood well, the first sentence is conclusion from review or previous studies, not from this one. If so, it is better specify with the "As per previous studies, ...":

236 These RTs were indicators of generalized individual differences, and variations within
237 individuals had been regarded as errors.

General comments:

Please, work tightly with a native speaker, but who understands the study also for a better Grammar but also the context description. For example, here, the subject has changed in the 2nd part of the sentence:

"201 The subjects included in cluster 1 had a decreased cerebral blood flow (93.3%), and the cluster 2 was
202 not change."

Small hints:

1) No need to put a dot after "p", here and further:

184 were observed between cluster (F(1, 30) = 48.28, p.<.01), and between BigFive Factors

2) Missing in coursive "(F(4, 30)=0.28, ns.)" in:

187 and between BigFive Factors (F(4, 30)=11.31, p.<.01) was not observed (F(4, 30)=0.28, ns.), and no

Author Response

Thank you for your comment. The table below has been revised according to the suggestive comments received from you. This amendment correspondence table is a sentence before sending it to MDPI English review service. There is also a place where expression was involved in the results of the English Review Service. We are sorry to trouble you, but we would really appreciate it if you could confirm. We really appreciate it.Our amendment table will be sent as an attached file.

Comments

Correction

Abstract:

1) Please, revise whether it is correct "obedient and obedient"

Thank you for your advice. The text has been corrected as follows:

The participants in cluster 2 were obedient and conservative.

2) In what the cluster 1 has decreased, please, specify since it is not clear:

Including comments from reviewer 2, the text has been corrected as follows:

Moreover, immediately after the RT measurement session with laptop personal computer (PC) started, the cluster 1 decreased HEG and increased amylase after the session and they were somewhat negative attitude.

May be the authored need to pass the correction or translation with a native speaker for it to improve.

We will use MDPI English editing service after correction.

METHODS

1) How the students were recruited, on which base (voluntary or any kind of compensation), and what were criteria of their inclusion/exclusion to the study?

All participants are Academic volunteers. However, we paid 1000JPY each participants as small reward. Added description as follows:

All participants were academic volunteers. We paid 1000 JPY each participants in slight reward.

2) Somewhere in the beginning it is good also to mention about Japanese "bipolar" scales since some readers may not know about it and imagine the Big Five they accustomed to use. It can be introduced here, for example, before "scale":

Changed "scale" to "bipolar scales".

3) Though some abbreviations are clear for many people, still, it is good to provide the disclosure of their abbreviations when it is mentioned for the first time in the manuscript: HEG, HR, EEG:

Added abbreviations description in parentheses.

RESULTS

1) Which "factor", please, specify: Or do you mean all factors?

In this ANOVA, we confirmed session 2 RT, which is the base variable for sRT derivation. If there was interactions between the stimulus term factor and the cluster, we planned to compare using the least square means of interaction. However, it showed no interaction, indicating that inter-individual differences could not be detected using RT in directly. The result from ANOVA show that Cluster 2 have longer RTs for personality ration, regardless of the stimulus term factor. The cluster analysis was performed using sRT according to Equation 1 in Fig. 1 as the classification variable. This analysis shows that RT cannot detect intraindividual differences, however sRT can detect intraindividual differences. It may be possible to delete it if it is not necessary due to misunderstandings in this description.

2) In Table 3, it will be better if you specify (or even give titles) to each cluster with reference on RT and/or other parameters

Added the following Footnote to Table 3, and added Bigfive factors to the Trait Word group.

Note: Each traits words were in accordance with Big Five in Table 1. The unit of RT(Reaction Time) were milliseconds. Also sRT were coefficients calculated as the ratio of RT by equation in Figure 1.

3) Please, specify, which "dependent variable" (line 193)?

It shows sRT, which is a classification variable for cluster analysis. The text has been corrected as follows:

As a result from the cluster analysis using sRT as classification variables,

4) Please, give the disclosure for abbreviations given on Figures 4 & 5 in the Notes below.

Added to Table 2. Added notes to Figures 4 and 5.

5) At the Figure 3, please, specify in Notes what the numbers in two vertical columns stand for? (Subjects ids and ???)

Added the following note.

Notes: The vertical axis were participant ID and corresponding number, and the horizontal axis was similarity index of clustered dendrogram.

6) Why the authors considered p<.10 as statistically significant level or there is a typing error:

Since the number of samples is small, the criterion for significant differences has become stricter. Therefore, for the sake of convenience, we allowed significant level of 10% for further research. As you indicated, the significance level of 10% was considered inappropriate as a description of the statistical test results. In a statistical test p<.10 changed to "ns.".   if it had the significant difference tendency, added the sentences.

7) The small sample size should be explained as Limitations of study.

As you indicated, we added the small sample issues to the discussion.

DISCUSSION

Study Limitations should be mentioned as sample size (though for physiological measurements less sample size is used compared to other psychological studies usually due to procedure). Also the direction for future studies it is good to indicate in the Discussion.

Thank you for kindly comments. We added following text and reference to the end of the discussion.

This study is based on HEG. This study should be replicated with other novel neuroimaging modality. Functional near-infrared spectroscopy (fNIRS) is a non-invasive neuroimaging technology that maps the functions of the cerebral cortex by measuring hemodynamics and demonstrates cost-effectiveness [14]. fNIRS has the advantage of translating to clinical use as it is more cost-effective than functional magnetic resonance imaging (fMRI) [15]. Future research should study RT by fNIRS in clinical populations.

14. Lai, C. Y., Ho, C. S., Lim, C. R., & Ho, R. C. (2017). Functional near-infrared spectroscopy in psychiatry. BJPsych Advances, 23(5), 324-330.

15. Ho, C. S., Zhang, M. W., & Ho, R. (2016). Optical Topography in psychiatry: A Chip Off the Old Block or a new Look Beyond the Mind–Brain Frontiers?. Frontiers in psychiatry, 7, 74.

CONCLUSIONS

If I understood well, the first sentence is conclusion from review or previous studies, not from this one. If so, it is better specify with the "As per previous studies, ...":

Thank you for your advice. We agree with your opinion, we rewrote the beginning of the sentence.

General comments

Please, work tightly with a native speaker, but who understands the study also for a better Grammar but also the context description. For example, here, the subject has changed in the 2nd part of the sentence:

Thank you for the advice. We corrected as follows. We will use MDPI review service.

The subjects included in cluster 1 had decreased cerebral blood flow (93.3%), and the subjects included in cluster 2 was not change.

Small hints:

1) No need to put a dot after "p", here and further:

Thank you for your suggestion. dot was omitted.

2) Missing in cursive "(F(4, 30)=0.28, ns.)" in:

Thank you for your advice. I fixed it.

Reviewer 2 Report

Thank you for inviting me to review the paper on “The classification of participants based on reaction time increase-decrease rate model on the personality traits word” This is an interesting study using HEG (near infrared hemoencephalography) and objective measures to assess personality. I think this paper is publishable after revision. I am happy to review the paper again. My recommendations are as follows:

Under the abstract, the authors stated, “The participants in cluster 2 were obedient and obedient.” The word obedient appeared twice. Please remove one of them. Under the abstract, the authors stated, “Moreover immediately after the PC session started”. What is PC? Under the introduction, the authors said, “Next, as the mental performance of self-rating personality, we measure the RT in two sessions; a self-rating term session and a self-rating by-sentence session.” What is “term” and “by-sentence” session? Please define. Can the authors provide the source or origin of the formula: Figure 1. The equation for the calculation of sRT? Is there a reference for this formula? Was this formula developed by the authors? For figure 4, which side of the graph is HEG values and which side is amylase levels? The authors need to label properly. Under discussion, the authors stated, “The personality inspection by the questionnaire was somewhat self-reporting [11]”. Please amend the statement as follows:

.. The personality inspection by the questionnaire was somewhat self-reporting [11] and a lack of internal consistency [Keng et al 2019].

Reference

Kent SL et al (2019) Construct Validity of the McLean Screening Instrument for Borderline Personality Disorder in Two Singaporean Samples J Pers Disord, 33 (4), 450-469

Under discussion, the authors should add a section of further research. I suggest using functional near-infrared spectroscopy to replicate this study. Please add the following statements:

This study is based on HEG. This study should be replicated with other novel neuroimaging modality. Functional near-infrared spectroscopy (fNIRS) is a non-invasive neuroimaging technology that maps the functions of the cerebral cortex by measuring hemodynamics and demonstrates cost-effectiveness (Lai et al. 2017). fNIRS has the advantage of translating to clinical use as it is more cost-effective than functional magnetic resonance imaging (fMRI) (Ho et al 2016). Future research should study RT by fNIRS in clinical populations.

References

Lai CY et al 2017. Functional near-infrared spectroscopy in psychiatry. BJPsych Advances September 2017, 324-330. 

Ho CS et al 2016. Optical Topography in Psychiatry: A Chip Off the Old Block or a New Look Beyond the Mind-Brain Frontiers? Front Psychiatry.7:74.

Author Response

Thank you for your comment. The table below has been revised according to the suggestive comments received from you.

This amendment correspondence table is a sentence before sending it to MDPI English review service. There is also a place where expression was involved in the results of the English Review Service. We are sorry to trouble you, but we would really appreciate it if you could confirm. We really appreciate it.

Comments

Correction

Thank you for inviting me to review the paper on “The classification of participants based on reaction time increase-decrease rate model on the personality traits word” This is an interesting study using HEG (near infrared hemoencephalography) and objective measures to assess personality. I think this paper is publishable after revision. I am happy to review the paper again. My recommendations are as follows:

Dear reviewer,

Thank you so much your very suggestive and important comments. Thank you for your specific comments on the manuscript. I will revise the manuscript according to your suggestions.

I will also using MDPI editorial service to proofread English expression.

Under the abstract, the authors stated, “The participants in cluster 2 were obedient and obedient.” The word obedient appeared twice. Please remove one of them.

Thank you for your advice. The text has been corrected as follows:

The participants in cluster 2 were obedient and conservative.

Under the abstract, the authors stated, “Moreover immediately after the PC session started”. What is PC?

Sorry, PC means laptop personal computer. Including comments from reviewer 1, the text has been corrected as follows:

Moreover, immediately after the RT measurement session with laptop personal computer (PC) started, the cluster 1 decreased HEG and increased amylase after the session and they were somewhat negative attitude.

Under the introduction, the authors said, “Next, as the mental performance of self-rating personality, we measure the RT in two sessions; a self-rating term session and a self-rating by-sentence session.” What is “term” and “by-sentence” session? Please define.

Thank you for your comments. The text has been corrected as follows:

We measured RT and cerebral blood flow during three rating sessions on the computer: a simple-response session, a self-rating react with words only by-term session, and a self-rating react with text by-sentence session.

Can the authors provide the source or origin of the formula: Figure 1. The equation for the calculation of sRT? Is there a reference for this formula? Was this formula developed by the authors?

Formula of sRT is the increasing rate of stimuli to reaction time model. The formula in Figure 1 is an empirical formula that detects individual differences from RT. It is our original. We added it to reference.

Sato, E., & Matsuda, K. (2016). The Feature of the Reaction Time for Performing Personality Self-rating: Conditions by Personality Trait Terms and by Sentence, Japanese journal of applied psychology, 42, 8-15.

For figure 4, which side of the graph is HEG values and which side is amylase levels? The authors need to label properly.

We added the following note to Figure 4.

Note: CL:Eyes closed for 3 minutes as a rest time. BF: Self-rating with BigFive Inventory before PC sessions. S1: Simple response to trait words session 1 on PC. S2: Personality self-rating by-term session 2 on PC. S3: Personality self-rating by-sentence session 3 on PC. QA: Self-rating with questionnaire after PC sessions. Pre: Salivary amylase before PC session. Post: Salivary amylase after all sessions.

Under discussion, the authors stated, “The personality inspection by the questionnaire was somewhat self-reporting [11]”. Please amend the statement as follows:

The personality inspection by the questionnaire was somewhat self-reporting [11] and a lack of internal consistency [Keng et al 2019].

Keng SL et al (2019) Construct Validity of the McLean Screening Instrument for Borderline Personality Disorder in Two Singaporean Samples J Pers Disord, 33 (4), 450-469

Thank you for your kind comments. We corrected it as you indicated.

The personality inspection by the questionnaire was somewhat self-reporting [12] and a lack of internal consistency [13].

13. Keng SL et al (2019) Construct Validity of the McLean Screening Instrument for Borderline Personality Disorder in Two Singaporean Samples J Pers Disord, 33 (4), 450-469

Under discussion, the authors should add a section of further research. I suggest using functional near-infrared spectroscopy to replicate this study. Please add the following statements:

This study is based on HEG. This study should be replicated with other novel neuroimaging modality. Functional near-infrared spectroscopy (fNIRS) is a non-invasive neuroimaging technology that maps the functions of the cerebral cortex by measuring hemodynamics and demonstrates cost-effectiveness (Lai et al. 2017). fNIRS has the advantage of translating to clinical use as it is more cost-effective than functional magnetic resonance imaging (fMRI) (Ho et al 2016). Future research should study RT by fNIRS in clinical populations.

Thank you for your kind comments and advices. We added it to the end of the discussion, as you indicated.

This study is based on HEG. This study should be replicated with other novel neuroimaging modality. Functional near-infrared spectroscopy (fNIRS) is a non-invasive neuroimaging technology that maps the functions of the cerebral cortex by measuring hemodynamics and demonstrates cost-effectiveness [14]. fNIRS has the advantage of translating to clinical use as it is more cost-effective than functional magnetic resonance imaging (fMRI) [15]. Future research should study RT by fNIRS in clinical populations.

14. Lai, C. Y., Ho, C. S., Lim, C. R., & Ho, R. C. (2017). Functional near-infrared spectroscopy in psychiatry. BJPsych Advances, 23(5), 324-330.

15. Ho, C. S., Zhang, M. W., & Ho, R. (2016). Optical Topography in psychiatry: A Chip Off the Old Block or a new Look Beyond the Mind–Brain Frontiers?. Frontiers in psychiatry, 7, 74.

Round 2

Reviewer 2 Report

.